# Geographical Distribution and Multimethod Species Identification of Forensically Important Necrophagous Flies on Hainan Island

**DOI:** 10.3390/insects14110898

**Published:** 2023-11-20

**Authors:** Yihong Qu, Bo Wang, Jianqiang Deng, Yakai Feng, Zhiyun Pi, Lipin Ren, Jifeng Cai

**Affiliations:** 1Department of Forensic Science, School of Basic Medical Sciences, Central South University, Changsha 410017, China; quyihong88@163.com (Y.Q.); cynthia090721@126.com (Z.P.); 2Hainan Equity Judicial Expertise Center, Hainan Vocational College of Political Science and Law, Haikou 570100, China; 3Hainan Provincial Academician Workstation, Haikou 570100, China; sapphirestrings@sina.com (B.W.); forensicpatho@163.com (J.D.); 4Department of Forensic Medicine, School of Basic Medical Sciences, Xinjiang Medical University, Urumqi 830011, China; fyk734@126.com; 5Shanghai Key Lab of Forensic Medicine, Key Lab of Forensic Science, Ministry of Justice, Academy of Forensic Science, Shanghai 570100, China

**Keywords:** forensic entomology, geographical distribution, species identification, Hainan Island

## Abstract

**Simple Summary:**

Forensic entomology offers unique advantages for the minimum postmortem interval (PMImin) estimation of decomposed corpses in forensic investigations. Molecular identification has become a routine and accurate tool. In this study, the community structure of the necrophagous flies on Hainan Island was investigated in detail according to the geographical environment. A GenBank database was preliminarily constructed to provide reference data for species identification in this area. We found that a high-resolution melt (HRM) curve analysis was a fast, cost-effective method for the species identification of flies. This study enriches the database of forensically important flies in tropical rainforest regions.

**Abstract:**

Forensic entomology offers unique advantages for the minimum postmortem interval (PMImin) estimation of decomposed corpses in forensic investigations. Accurate species identification and up-to-date locality information are essential. Hainan Island has a tropical rainforest climate and a vast territory. In this study, the community structure of necrophagous flies on Hainan Island was investigated in detail according to geographical environment. The results showed that the dominant species included *C. megacephala*, *S. peregrina*, *C. rufifacies*, *S. misera*, *H. ligurriens*, *S. sericea*, *S. cinerea*, *S. dux*, *C. pinguis*, and *M. domestica*. Furthermore, *C. rufifacies* and *C. villeneuvi* were found only in the high-altitude areas of Wuzhi Mountain, while *S. cinerea* was distributed only in coastal areas; the latter is a representative species of Hainan Island and has not been reported before. Furthermore, a GenBank database of forensically important flies was established, whilst a high-resolution melt (HRM) curve analysis was applied to identify the common species of Hainan Island for the first time. This study enriches the database of forensically important flies in tropical rainforest regions.

## 1. Introduction

Forensic entomology mainly focuses on the presence of insects and other arthropods in forensic investigations. The age determination of necrophagous flies is one of the key steps used to provide evidence for a minimum postmortem interval (PMImin) estimation [1,2]. Generally, the developmental patterns of immature insects take place in a predictable manner under a controlled temperature, but insect development is species-specific [3,4]; more specifically, the timing of each life stage at any given temperature is species-specific, which is why correct species identification is so important. As there are significant differences in the geographical distribution of insects, accurate species identification and up-to-date locality information are essential for the effective application of forensic entomology in criminal investigations [5].

A survey of the distribution of necrophagous flies is an effective way to understand their population and biological characteristics [6]. The community structure of necrophagous flies varies with the geographical region, habitat, vegetation, soil type, and meteorological conditions, which have significant impacts on the geographical distribution of insects [7,8,9]. Located in the southernmost part of China, Hainan is a tropical island with a unique environment and high temperatures. It is the province with the longest coastline in China and it has a mountainous area in the middle. The annual average temperature of Hainan Island is 22 °C–26 °C, and it is lower in the central mountainous area and higher in the southwest. Hainan has a large area and a diverse natural environment, but there is little video surveillance, so corpses are usually found in an advanced stage of decomposition. Forensic entomology can provide effective evidence in such cases.

The species identification of insects is an important task in forensic entomology [5]. One of the challenges of modern biology is to develop accurate and reliable methods to rapidly identify species. Although the method of morphological identification has the advantage of low cost, some forensically important flies are morphologically difficult to distinguish, especially at the juvenile stage, and preservation by the crime scene technician makes rearing them to the adult stage impossible [10,11]. In recent years, a large number of studies have shown that the use of molecular data is one of the fastest and most reliable methods for species identification. Further, molecular identification can solve these problems, especially for scientists without formal taxonomic training, and it can be applied to ancient or damaged samples [10]. COI barcodes have been mainly used for the molecular identification of forensically important flies [11], such as 651 bp of the COI gene, which were applied to identify 42 forensically important Diptera species, indicating that they are suitable markers for the species identification of necrophagous flies in Spain [12]. Combined sequences (2216 to 2218 bp) of the cytochrome c oxidase subunit I and II genes can serve as reliable reference databases to identify the 14 forensically important flesh fly species in Thailand [13]. Jafari et al. obtained sequence data from the COI-COII regions for 10 flesh fly species collected in Iran, providing distinct restriction fragment length polymorphism profiles among the species [14]. Bharti and Singh developed COI reference data for nine blow fly species collected in India, but *Chrysomya megacephala* and *Chrysomya chani* could not be distinguished reliably [15]. Although these molecular markers may serve as identification tools for forensically important flies, the genetic sequences available in GenBank are still limited, especially in Hainan Island. In this study, a GenBank database of common necrophagous flies was constructed in detail for the first time, which provides an important data reference for the study of necrophagous insects not only in Hainan Island, but also in other tropical regions.

Additionally, a high-resolution melt (HRM) curve analysis can provide single-base-pair differentiation. HRM is a closed-tube and post-PCR method that can analyze the genetic variation in PCR amplicons, and since different gene sequences melt at slightly different rates, subtle differences in the melting temperature (Tm) can be detected by melting curves [16]. An HRM curve analysis is a fast and cost-effective tool that allows distinguishing of differences between PCR products, even as little as a single base pair. HRM does not require fluorescently labeled probes or separation steps and has already been used in single-nucleotide polymorphism (SNP) typing mutation detection. Depending on the type of SNP, the difference in Tm can vary from 0.1 °C to 1.5 °C between the two homozygous alleles [17]. An analysis of HRM curves has been applied successfully in molecular diagnostics to distinguish microorganisms [18]. Furthermore, Kang and Sim proposed a novel identification method using an HRM analysis by examining single-nucleotide polymorphisms in the acetylcholinesterase-2 (ace-2) locus, providing a high confidence for species determination among the three Culex complex mosquitoes [19]. The HRM approach can serve as an alternative tool to DNA barcoding for the taxonomic identification of invertebrates, as shown by Ajamma et al., who verified the differentiation of Anopheles species using a PCR-HRM analysis and found that the COI-barcode-region primers had the best and most definitive effect on the separation of mosquito species [20]. Additionally, a highly specific and sensitive HRM approach was developed to accurately identify *Drosophila suzukii* samples [21]. At present, an HRM curve assay is used, developed to identify eight earthworm species common to agricultural soils in Central Europe [22]. Therefore, in contrast to DNA barcoding, an HRM curve analysis to distinguish PCR products based on their thermal denaturation properties is a fast and cost-effective post-PCR tool for identifying taxa. It has, however, rarely been used in the species identification of forensically important flies.

On the basis of a previous preliminary study [23], in order to further supplement the reference data available for tropical forensic entomology, in this study, the species distribution of necrophagous flies in Hainan Island was investigated in detail according to the natural environment, and then a GenBank database was established. Furthermore, we demonstrated the potential of an HRM curve analysis of the COI gene to distinguish among necrophagous fly species, and we developed a rapid and inexpensive method to reduce the cost of molecular identification.

## 2. Materials and Methods

### 2.1. Specimen Collection

Due to the unique geographical environment and tropical monsoon climate of Hainan Island, it can be divided into the following regional climate standards according to the temperature, humidity, light, and precipitation conditions: semi-humid, humid, semi-arid, mountainous humid, and semi-arid/humid. In order to comprehensively reflect the diversity of species distribution, three different natural environments (urban, forest, and coastal) were selected as sampling sites in each climate zone (Figure 1 and Table 1). Due to the lack of obvious seasonal differentiation in Hainan Island, we selected unique sampling times, which were classified into a dry season (November to February, low temperature and little precipitation) and a rainy season (March to October, high temperature and more precipitation) according to temperature and precipitation.

In this study, the sampling period was from 1 December 2020 to 30 November 2021. Adult specimens were collected from three traps at each sampling site. About 200 g of pig lungs was placed in each trap at 7 o’clock every day, and the samples were collected at 18 o’clock. Each sampling site was collected continuously for 7 days. A total of 15 sampling sites were selected according to the geographical location of Hainan Island. All specimens were killed by freezing and then identified through traditional morphological keys, which was performed by a forensic entomologist (Lushi Chen) [24]. The samples were stored in the Cai lab (Haikou, Hainan, China).

### 2.2. DNA Extraction and Sequencing

Based on the morphological identification, some widely distributed species were selected, and then the COI genes of these species were sequenced to establish a gene bank of common necrophagous flies in Hainan Island. These samples were originally preserved in a 95% ethanol, and then transferred to ~20 °C conditions. All voucher specimens were assigned a unique field code and deposited in Guo’s lab. Specifically, a total of 30 species were chosen for further analysis, and a specimen was chosen for each species. The total genomic DNA was extracted from the thorax muscle tissues of each specimen using the QIANamp Micro DNA Kit (Qiagen BIOTECH CO., LTD, Dusseldorf, Germany) according to the manufacturer’s protocol. The COI gene was amplified using the following designed primer pairs: TY-J-1460: 5′-TACAATTTATCGCCTAAACTTCAGCC-3′ and C1-N-2800: 5′-CATTTCAAGCTGTGTAAGCATC-3′. The PCRs were carried out with TaKaRa LA Taq (TaKaRa, Dalian, China) under the following conditions: 5 min of initial denaturation at 94 °C; 35 cycles of 60 s at 94 °C, annealing at 50 °C for 30 s, and 6 min at 72 °C; and a final elongation for 10 min at 72 °C. Each 20 μL PCR reaction contained 10 μL of PCR buffer, 1 μL of each 10 μM primer, 2 μL of the DNA template, and nuclease-free water. The PCR products were separated on a 1.0% agarose gel, visualized using ethidium bromide staining, and then purified using the AidQuick Gel Extraction Kit (Qiagen, Germantown, MD, USA). Sequencing was performed directly using an ABI PRISM 3730 automated sequencer (Applied Biosystems, Foster, CA, USA). Sequence alignments were performed using MAFFT v7.520 [25]. A phylogenetic analysis was carried out using the neighbor-joining (NJ) method. Phylogenetic analyses and determinations of genetic divergence were performed by utilizing the uncorrected pairwise p-distance model in MEGA X [26]. A bootstrap consensus tree was generated with 500 bootstrap replicates.

### 2.3. High-Resolution Melting (HRM) Curve Analysis

The COI gene was used as a target in designing the PCR amplicon. Reference sequences of eight dominant species from Hainan Island were obtained from the sequencing data mentioned above, three specimens were chosen for each species, and then suitable primers were designed as follows: forward C1-J-2495: 5′-CAGCTACTTTATGAGCTTTAGG-3′ and reverse C1-N-2800: 5′-CATTTCAAGCTGTGTAAGCATC-3′. The length of the amplified product was 278 bp. PCR primers were purchased from Tsingke Biotech (Beijing, China). Each PCR reaction mix was prepared using 10 μL of MonAmp^TM^ 2×Taq mix Pro, 2 μL of each 10 μM primer, 1 μL of 20×Eva Green, 2 μL of DNA, and 5 μL of H2O for a total volume of 20 μL. PCR cycling and an HRM analysis were performed on the Rotor-Gene Q (Qiagen, Dusseldorf, Germany). The thermocycling conditions were as follows: 10 min of initial denaturation at 94 °C; 35 cycles of denaturation for 30 s at 94 °C, annealing at 54 °C for 30 s, and elongation for 6 min at 72 °C; and a final elongation for 10 min at 72 °C. The melting curve data were generated by increasing the temperature from 60 °C to 95 °C at 0.1 °C per second and recording the fluorescence. The HRM curve analysis was performed using the Rotor-Gene 1.7.27 software, and the HRM algorithm was provided. After the HRM, the PCR products were separated on a 2.0% agarose gel and imaged using a Tanon 3500R gel-imaging device (Tanon, Hangzhou, China) to observe the brightness of the bands.

## 3. Results

### 3.1. Species Richness and Diversity

According to the morphological classification of necrophagous flies, a total of 12,251 flies were obtained from 5 families, 17 genera, and 36 species in Hainan Island, among which 12 species were relatively common. These species were *Chrysomya megacephala* (3400), *Sarcophaga peregrina* (1346), *Chrysomya rufifacies* (1131), *Sarcophaga misera* (1008), *Hemipyrellia ligurriens* (980), *Sarcophaga sericea* (700), *Sarcophaga cinerea* (630), *Sarcophaga dux* (562), *Chrysomya pinguis* (536), *Musca domestica* (416), *Lucilia porphyrina* (205), and *Chrysomya villeneuvi* (200). A total of 11,114 specimens were captured, which accounted for 90.72% of the total samples captured on Hainan Island (Appendix A). The results further showed that *Lucilia cuprina* can be collected in urban areas and on forest land, but not in coastal areas, and *Lucilia porphyrina* was only collected on forest land. *Sarcophaga scopariiformis* and *Sarcophaga caudagalli* were found only in urban areas, and *Sarcophaga cinerea* was only collected by the sea. *Synthesiomyia nudiseta* was more prevalent in the urban and coastal areas than in the woodland.

The highest number of necrophagous flies were collected from the semi-humid area (A), followed by the semi-arid area (C), semi-arid/-humid area (E), mountainous humid area (D), and humid area (B). Calliphoridae and Sarcophagidae were found in almost equal numbers in the semi-humid area, humid area, semi-arid area, and semi-arid/-humid area, but Calliphoridae accounted for 73.362% in the mountainous humid area, and a few Anthomyidae specimens were found in the humid area and the mountainous humid area (Figure 2a). At the genus level, *Lucilia* was significantly more prevalent in the humid mountain areas than in the other climate zones (Appendix A). In terms of species-level classification, the four climate zones were generally similar, except for large differences in the species structure in the mountainous humid zone (Figure 2b). We further determined that *Chrysomya pinguis*, *Achoetandrus villeneuvi*, and *Lucilia porphyrina* were only found in the mountainous humid areas. *Sarcophaga similis* was only found in the semi-humid areas and mountainous humid areas, and *Sarcophaga formosensis* was also collected in the humid areas. *Seniorwhitea reciproca* was found in all zones.

### 3.2. COI for Species Identification

This study showed that the molecular identification was consistent with the results of the morphological classification. At the family level, the three families (Sarcophagidae, Calliphoridae, and Muscidae) were well distinguished, and Anthomyidae was separated independently as an outgroup. The clustering analysis, as expected, showed the division of samples into three clades, and the bootstrap values were higher than 70% (Figure 3 and Appendix A). In Sarcophidae, *S. misera*, *S. albiceps*, *S. brevicornis*, and *S. dux* were closely related and clustered into a branch as a subgenus. *S. peregrina* and *S. formosensis* belong to the subgenus of *Boettcherisca* and could not be clearly distinguished. *S. scopariiformis* also gathered together with *S. peregrina* and *S. formosensis*, but with obvious branches. These results suggest that *S. scopariiformis* may have diverged from the subgenus *Boettcherisca*. *S. caudagalli* and *S. pattoni* were clustered together and had separate branches. *S. princeps*, *S. ruficornis*, and *S. cinerea* were sisters to each other, and *S. princeps* and *S. ruficornis* were more closely related. *S. josephi* was in a separate clade with an independent evolutionary relationship with other species. In Calliphoridae, the species of *C. megacephala*, *C. pinguis*, *C. villeneuvi*, and *C. rufifacies* were sisters to each other, and were clustered into an independent branch of the genus *Chrysomya*. *L. hainanensis* and *L. porphyrina*, *L. cuprina* and *L. sericata*, and *L. bazini* and *H. ligurriens* were sisters to each other, and all belonged to the genus *Lucilia*. In Muscidae, *H. chalcogaster* was closely related to *H. spinigera*. *S. nudiseta*, as an independent branch, was a sister to both *M. sorbens* and *M. domestica*, which were also sisters to each other. As a separate branch, *M. stabulans* was a sister to other species of *Musca* and *Hydrotaea*.

Based on the above phylogenetic analysis, the interspecific divergence was further calculated by computing the pairwise distance. The results show that in Sarcophidae, except for the interspecific divergence of *S. peregrina* and *S. formosensis*, which was 2.8%, the remaining species’ values, ranged from 4.8% to 11.2%. In Calliphoridae, the interspecific divergences of *C. megacephala* and *C. pinguis* and of *L. cuprina* and *L. sericata* were 2.5% and 1.0%, respectively, and all other species were in the range of 5.5–9.9%. Within Muscidae, the interspecific divergence ranged from 6.9% to 12.5%. Based on the above results, the COI sequences of the common necrophagous flies on Hainan Island were submitted to GenBank (accession No. OQ519752-OQ519781) (Table 2).

### 3.3. Molecular Identification Based on HRM Analysis

According to the length of the DNA, the GC content, and the base complementarity, the samples were analyzed using HRM curves, in which a high temperature uniformity and temperature resolution enabled the theoretical resolution to reach the distinction of single-base polymorphisms. In this study, eight dominant species were selected for HRM analysis. A single characteristic peak was obtained for all species except *L. porphyrina*, and the Tm values of different species were separated. Peak values were 79.62 for *H. ligurriens*, 76.83 for *S. peregrina*, 73.35 for *C. megacephala*, 76.60 for *S. dux*, 76.35 for *S. misera*, 77.23 for *S. sericea*, and 73.13 for *C. pinguis*, and the corresponding Tm values were 73.15, 76.1, and 79.01 for *L. porphyrina* (Figure 4 and Figure 5). In order to further verify the effectiveness of this HRM system for species identification, test species (three samples) were selected after the DNA was extracted, and the established system was applied for the HRM analysis (Figure 6). The results showed a single peak curve with a Tm value of 76.73, which differed from *S. peregrina* by only 0.1 °C and had a unique corresponding relationship. 

## 4. Discussion

This study investigated the community structure of necrophagous flies on Hainan Island. The dominant necrophagous flies on Hainan Island were *C. megacephala*, *S. peregrina*, *C. rufifacies*, *S. misera*, *H. ligurriens*, *S. sericea*, *S. cinerea*, *S. dux*, *C. pinguis*, *M. domestica*, *L. porphyrina*, and *C. villeneuvi*. Among them, *S. cinerea* was only found in coastal areas. The seasonal distribution of necrophagous flies was mainly related to temperature [27,28]. Hainan Island is located in the tropics and has high temperatures all year round. A large number of *L. porphyrina* were collected from the Wuzhi Mountain sampling site, with an altitude of 1331 m. Furthermore, *C. megacephala* was distributed in all sampling sites, which may be due to its tolerance of a wide range of temperatures, resulting in it being active in temperate regions from May to December. Hainan Island has a tropical oceanic monsoon climate, and the precipitation is closely related to the wind direction. The majority of the precipitation in summer comes from the southwest monsoon off the Indian Ocean, and in winter it comes from the northeast monsoon off the mainland. However, the special topography of Hainan Island is “high in the middle and low around”, which blocks the monsoon from two directions. Therefore, in this study, Hainan Island was divided into five climate zones for sampling. It was found that the fly species were similar in the arid and semi-arid/-humid regions, followed by the semi-humid regions, mountainous humid regions, and humid regions. In addition, we also found that *S. cinerea* was the endemic fly species in the coastal areas of Hainan Island.

To date, DNA barcoding has been established as a reliable method for the identification of diptera species, such as those belonging to Calliphoridae, Sarcophagidae, Phoridae, and Muscidae [13,29,30,31,32]. Molecular markers have unique advantages in the identification of closely related species; for example, COI barcodes have been used to distinguish the species of the *Chrysomya* genus from the eastern part of Australia [33]. Therefore, in this study, we investigated the species distribution characteristics of common necrophagous flies found on Hainan Island in detail, and then the common species were selected for DNA extraction and sequencing. The results showed that COI can be used for species identification except in cases of the interspecific divergence of several species with values of less than 3%. For example, the phylogenetic difference between *S. peregrina* and *S. formosensis* is 2.8%, but their sister grouping has been demonstrated [34,35]. In Calliphoridae, the interspecific divergence between *C. megacephala* and *C. pinguis* is 2.5%. Nelson et al. reported that the interspecific variation within the *Chrysomya* genus is less than 0.5% [33]. The interspecific divergence between *L. cuprina* and *L. sericata* is 1.0%, and they are sister species, which makes it difficult to identify these species through COI barcoding; therefore, it is still necessary to find new genetic markers for species identification [36]. However, the sequence data are based on a single specimen for each species, and as such, the accuracy of the estimates of interspecies variation are limited. The validity of a genetic marker depends not only on the interspecific variation, but also on the intraspecific variation (particularly when dealing with closely related species, where there may only be approximately a 1% difference between the species, but a 0.5% difference within a species. Therefore, further investigation to reinforce the suitability of this genetic marker in more species is required, particularly in closely related species.

Although sequencing is still necessary in molecular identification to reveal the genetic diversity of haplotypes, the number of differentiated haplotypes based on HRM curves can serve as a first indicator of haplotype diversity. At present, HRM analysis has already been used for the species identification of diverse mosquitoes, spotted-wing drosophila, and earthworms [19,20,21], and it is a fast, cost-effective, and simple tool compared with sequencing technology. The feature of rapid detection is its most significant advantage [19]. In this study, the amplified fragment was less than 300 bp, which could be well detected, even for degraded samples. In addition, the sequencing technology has the disadvantages of being time-consuming and high in cost. However, in the process of HRM testing, the entire HRM analysis was performed within only 2–3 h. HRM can be conducted directly after PCR amplification, which is low cost, high throughput, and fast. Experimental detection only needs to design PCR primers, which are not limited by detection sites [37,38]. Furthermore, an HRM analysis only requires the detection of the intensity of fluorescence in the samples, which does not affect DNA after detection, and both amplification and detection are carried out in the same centrifuge tube, which avoids contamination.

In summary, the HRM technique was applied for the species identification of forensically important flies. Based on the dominant species collected on Hainan Island, the HRM method was preliminarily established, providing a reference for the development of the molecular identification of necrophagous flies. Therefore, we suggest that an HRM curve analysis could support morphological identification, especially for larvae and damaged specimens that cannot be identified morphologically. If morphology and molecular identification are combined, an HRM curve analysis may be the most suitable alternative to DNA barcoding, but this study requires a further increase in the number of samples for verification. Meanwhile, if specific primers with short amplified fragments can be designed for the degraded samples of more species in subsequent studies, the efficacy of the HRM analysis will be significantly improved. In addition, this study only focused on the COI gene, and the results may be further enhanced if nuclear genes can be combined.

## 5. Conclusions

This study investigated the species distribution characteristics of common necrophagous flies on Hainan Island in detail according to different climatic zones and natural environments, indicating that the community structure of necrophagous flies is related to the regional habitat, which enriches the database of forensically important flies in tropical rainforest regions. Moreover, a GenBank database was preliminarily established to provide genetic data for the species identification of forensically important flies in this region. But there are a large number of species in the database that are represented only by a single sequence; therefore, investigation is further required to strengthen the applicability of this marker to more species, especially in closely related species. Additionally, we found that an HRM curve analysis has the potential to identify more species simultaneously, and it is fast and cost effective for species identification, but the accuracy and efficacy of the method still need to be further verified with more samples.

## Figures and Tables

**Figure 1 insects-14-00898-f001:**
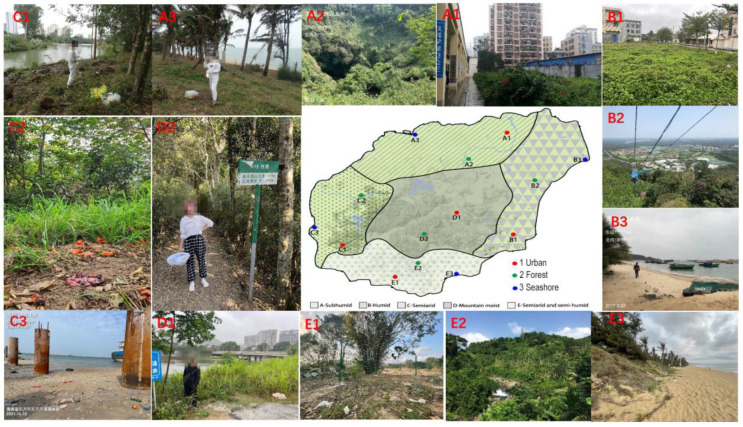
Geographical location and elevation of the experimental sites on Hainan Island, located in the southernmost part of China. This information is detailed in Table 1.

**Figure 2 insects-14-00898-f002:**
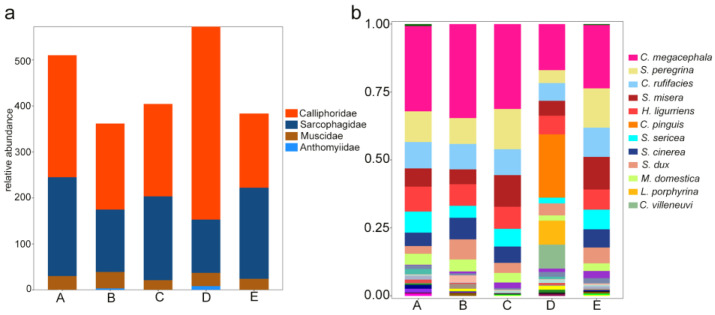
Relative abundance of necrophagous flies in different climatic regions of Hainan Island. (**a**) Relative abundance at the family level, where the labeled value indicates the number of species. (**b**) Relative abundance at the species level. A. Semi-humid zone; B. Humid zone; C. Semi-arid region; D. Mountainous humid area; and E. Semi-arid and semi-humid area.

**Figure 3 insects-14-00898-f003:**
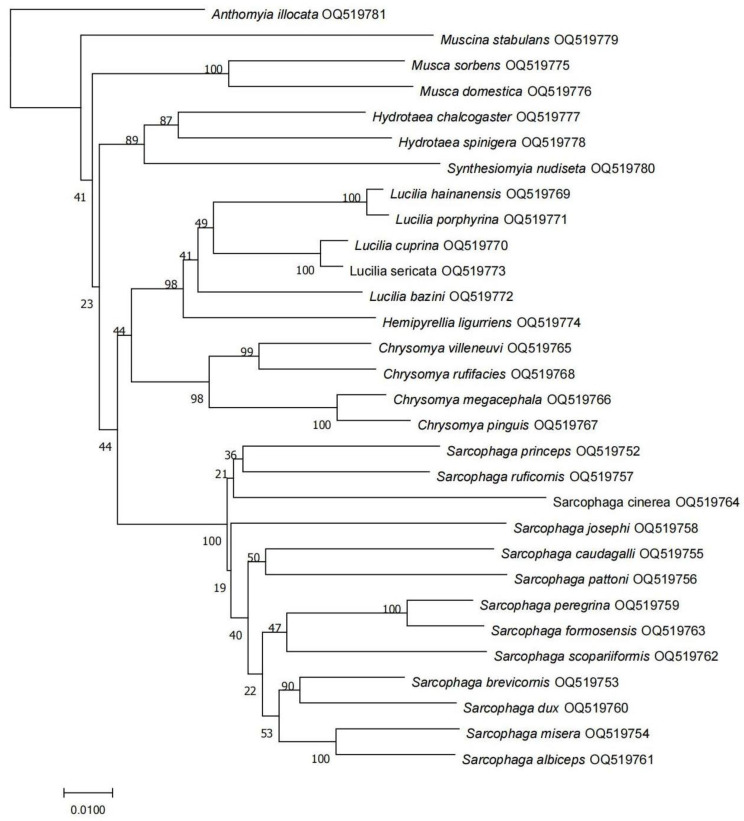
NJ tree of pairwise distances for COI sequences from 30 species. The morphological species identification and NCBI number are given in the specimen label. Numbers on branches indicate the bootstrap support value.

**Figure 4 insects-14-00898-f004:**
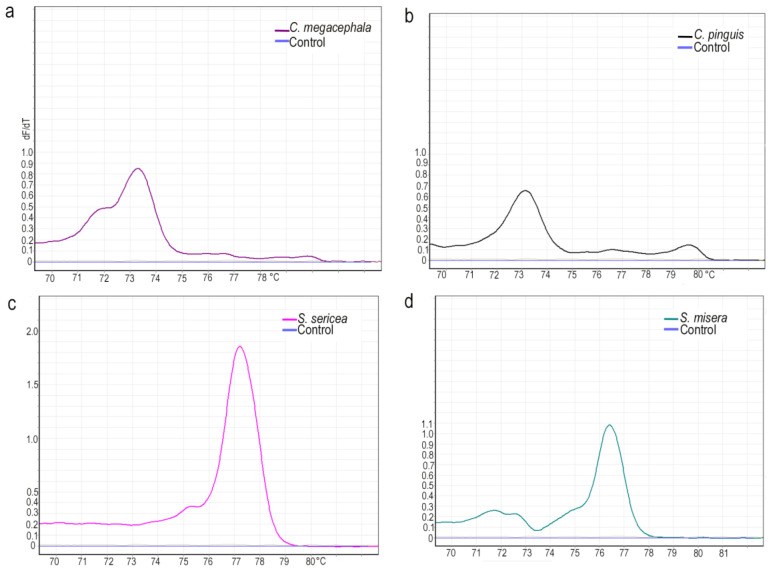
The specimens were analyzed using HRM curves. Different curve colors indicate different species. (**a**) Chrysomya megacephala, (**b**) Chrysomya pinguis, (**c**) Sarcophaga sericea, and (**d**) Sarcophaga misera.

**Figure 5 insects-14-00898-f005:**
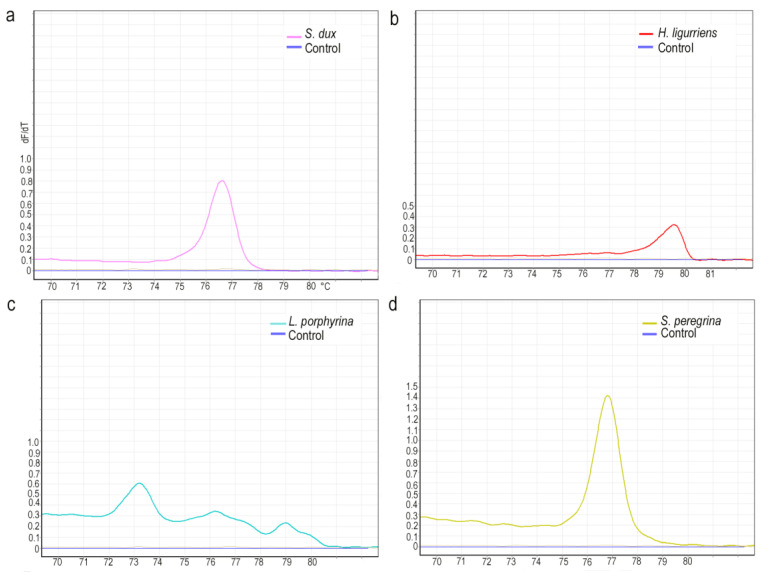
The specimens were analyzed using HRM curves. Different curve colors indicate different species. (**a**) Sarcophaga dux, (**b**) Hemipyrellia ligurriens, (**c**) Lucilia porphyrina, and (**d**) Sarcophaga peregrina.

**Figure 6 insects-14-00898-f006:**
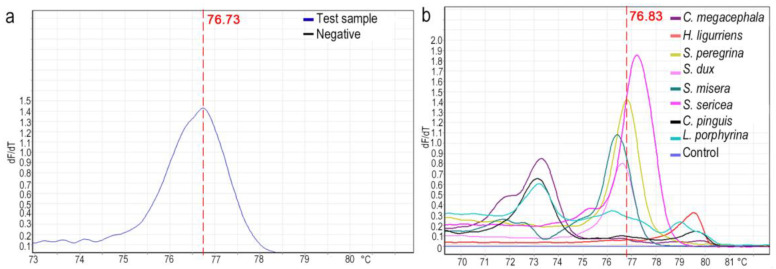
Validation of HRM evaluation system. (**a**) Melting curves of unknown species for verification. (**b**) HRM analysis based on 8 common species of flies.

**Table 1 insects-14-00898-t001:** Meteorological information of the geographical areas of Hainan Island.

Climate Zones	Site	Location	Environment	Latitude/Longitude	Altitude (m)	Average Temperature (°C)
Semi-humid	A1	North of Hainan Island	Urban	110°21′6″/19°58′48″	20.4	24
	A2		Forest	110°13′6″/19°55′32″	203.5	
	A3		Coastal	109°45′44″/19°57′52″	8.6	
Humid	B1	East of Hainan Island	Urban	110°22′48″/18°47′53″	35.2	24
	B2		Forest	110°27′20″/19°14′53″	198.3	
	B3		Coastal	110°1′44″/19°38′11″	2.4	
Semi-arid	C1	Southwest of Hainan Island	Urban	109°10′53″/18°45′38″	155.8	25
	C2		Forest	109°3′16″/19°18′17″	195.4	
	C3		Coastal	108°38′24″/19°6′36″	0.1	
Mountainous humid	D1	Center of Hainan Island	Urban	109°50′37″/19°1′56″	228.6	22.4
	D2		Forest	109°41′49″/18°53′42″	1331.3	
Semi-arid and semi-humid	E1	South of Hainan Island	Urban	109°36′19″/18°16′50″	23.8	25.7
	E2		Forest	109°26′15″/18°29′58″	187.5	
	E3		Coastal	109°44′28″/18°20′12″	8.8	

**Table 2 insects-14-00898-t002:** Collection location and reference data of adult specimens used for COI analysis in species identification.

No.	Species	Accession No. ^a^	Collected Location
1	*Sarcophaga princeps* (Wiedemann, 1830)	OQ519752	Wuzhishan (109°41′49″, 18°53′42″)
2	*Sarcophaga brevicornis* (Ho, 1934)	OQ519753	Sanya (109°44′28″, 18°20′12″)
3	*Sarcophaga misera* Walker, 1849	OQ519754	Sanya (109°44′28″, 18°20′12″)
4	*Sarcophaga caudagalli* Bottcher, 1912	OQ519755	Sanya (109°44′28″, 18°20′12″)
5	*Sarcophaga pattoni* Senior-White, 1924	OQ519756	Dongfang (108°38′24″, 19°6′36″)
6	*Sarcophaga ruficornis* (Fabricius, 1794)	OQ519757	Wanning (110°22′48″, 18°47′53″)
7	*Sarcophaga josephi* Bottcher, 1912	OQ519758	Haikou (110°21′6″, 19°58′48″)
8	*Sarcophaga peregrina* (Robineau-Desvoidy, 1830)	OQ519759	Sanya (109°44′28″, 18°20′12″)
9	*Sarcophaga dux* Thomson, 1869	OQ519760	Haikou (110°21′6″, 19°58′48″)
10	*Sarcophaga albiceps* Meigen, 1826	OQ519761	Wuzhishan (109°41′49″, 18°53′42″)
11	*Sarcophaga scopariiformis* Senior-White, 1927	OQ519762	Sanya (109°44′28″, 18°20′12″)
12	*Sarcophaga formosensis* (Kirneret-Lopes, 1961)	OQ519763	Haikou (110°21′6″, 19°58′48″)
13	*Sarcophaga cinerea* (Fabricius, 1794)	OQ519764	Wanning (110°22′48″, 18°47′53″)
14	*Chrysomya villeneuvi* Patton, 1922	OQ519765	Wuzhishan (109°41′49″, 18°53′42″)
15	*Chrysomya megacephala* (Fabricius, 1794)	OQ519766	Dongfang (108°38′24″, 19°6′36″)
16	*Chrysomya pinguis* (Walker, 1858)	OQ519767	Wuzhishan (109°41′49″, 18°53′42″)
17	*Chrysomya rufifacies* (Macquart, 1843)	OQ519768	Haikou (110°21′6″, 19°58′48″)
18	*Lucilia hainanensis* Fan, 1965	OQ519769	Haikou (110°21′6″, 19°58′48″)
19	*Lucilia cuprina* (Wiedemann, 1830)	OQ519770	Dongfang (108°38′24″, 19°6′36″)
20	*Lucilia porphyrina* (Walker, 1856)	OQ519771	Wuzhishan (109°41′49″, 18°53′42″)
21	*Lucilia bazini* Seguy, 1934	OQ519772	Dongfang (108°38′24″, 19°6′36″)
22	*Lucilia sericata* (Meigen, 1826)	OQ519773	Dongfang (108°38′24″, 19°6′36″)
23	*Hemipyrellia ligurriens* (Wiedemann, 1830)	OQ519774	Haikou (110°21′6″, 19°58′48″)
24	*Musca sorbens* Wiedemann, 1830	OQ519775	Wanning (110°22′48″, 18°47′53″)
25	*Musca domestica* Linnaeus, 1758	OQ519776	Sanya (109°44′28″, 18°20′12″)
26	*Hydrotaea chalcogaster* (Wiedemann, 1824)	OQ519777	Sanya (109°44′28″, 18°20′12″)
27	*Hydrotaea spinigera* Hennig, 1962	OQ519778	Haikou (110°21′6″, 19°58′48″)
28	*Muscina stabulans* (Fallen, 1817)	OQ519779	Dongfang (108°38′24″, 19°6′36″)
29	*Synthesiomyia nudiseta* (Wulp, 1883)	OQ519780	Wuzhishan (109°41′49″, 18°53′42″)
30	*Anthomyia illocata* Walker, 1856	OQ519781	Haikou (110°21′6″, 19°58′48″)

^a^ GenBank database accession number. The length of the COI sequence for all species was 1299 bp.

## Data Availability

The raw COI sequencing data were deposited in the database of the NCBI (National Center for Biotechnology Information) with accession numbers OQ519752-OQ519781.

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
