# Peer review of "Geographical Distribution and Multimethod Species Identification of Forensically Important Necrophagous Flies on Hainan Island"

_insects, 2023, doi:10.3390/insects14110898_

Round 1

Reviewer 1 Report (Previous Reviewer 1)

Comments and Suggestions for Authors

The manuscript has already improved considerably as a result of the revision, but the English in particular should be revised by a native speaker.

In addition, the discussion and conclusion still need improvement and should be revised in collaboration with an expert in the field of forensic entomology.

Comments on the Quality of English Language

The quality of the English could be improved, as the sentences are not well connected in some places, which makes reading difficult.

Author Response

Reviewer 1

1.The manuscript has already improved considerably as a result of the revision, but the English in particular should be revised by a native speaker.

Answer: Thank you very much for your recognition of our study. The comments were extremely helpful to improve the quality of the manuscript. The English has been revised by a native speaker.

2.In addition, the discussion and conclusion still need improvement and should be revised in collaboration with an expert in the field of forensic entomology.

Answer: We have tried our best to restate all suspicious issues in the revised manuscript. The revised parts have been highlighted and edited with track changes. We hope that the changes would fulfill the request for publication.

All the authors of this paper have agreed the finally revised manuscript. We wish to take this opportunity to thank you for consideration of publishing our paper.

Once again, thank you for your suggestions and comments.

Reviewer 2 Report (Previous Reviewer 2)

Comments and Suggestions for Authors

Edits in this new draft have much improved the manuscript with the inclusion of more background on the estimation of minimum PMI and the practical application of entomology in forensic science. The methodology is clearer with details of the sampling and PCR set up now included. Figure resolution has been improved.

Issues not yet addressed:

I'm concerned that the sequence data is based on a single specimen for each species and as such the accuracy of estimates of inter species variation are limited, and this limitation should be acknowledged. The validity of a marker for species ID depends not only on the variation between species but also the variation within a species (particularly when dealing with closely related species where there may only be approx. 1% difference between the species but maybe 0.5% difference within a species (ideally extensive population data would ensure markers chosen for ID can resolve differences between closely related species). It would be have been useful to include intraspecies variation within Table S2 (on the diagonal) to give a fuller picture of variation within COI or if the data are not available at least give some indication in the text of the general intraspecies variation seen in COI.

The HRM analysis is based on 3 specimens, it is still unclear how reproducible the peak Tms are. Is the value reported an average temperature from the three specimens if so the standard deviation around this would be important especially as some Tms for different species are quite close, showing the variation in measurements would add support to this being a novel and appropriate method of species ID. It would also be useful to understand the sequence divergence across this 300 bp target (i.e. how many nucleotide differences are causing this change in Tm).

Minor edits/typos:

Line 52/53 - you say insect development is species specific, more specifically it it the timing of each life stage at any given temperature that is species specific which is why correctly species identification is so important. This could be more clearly stated here.

Line 54 - Change "Whilst" to As there are significant differences in the geographical......

Good to see the Genbank accession numbers for the specimens sequences but this may be better placed in the supplementary data (could just refer to OQ519752 - 81 in the main text).

Supplementary data - not sure Figure S1 really adds anything, Table S2 I think would be better placed in the main body of the manuscript (add the length of COI sequence in the figure legend- is it the full length gene or just an amplified section?).

Comments on the Quality of English Language

The manuscript has been much improved in terms of the written English 

Author Response

Reviewer 2

Edits in this new draft have much improved the manuscript with the inclusion of more background on the estimation of minimum PMI and the practical application of entomology in forensic science. The methodology is clearer with details of the sampling and PCR set up now included. Figure resolution has been improved.

Answer: Thank you very much for your recognition of our study. The comments were extremely helpful to improve the quality of the manuscript.

Issues not yet addressed:

  1. I'm concerned that the sequence data is based on a single specimen for each species and as such the accuracy of estimates of inter species variation are limited, and this limitation should be acknowledged. The validity of a marker for species ID depends not only on the variation between species but also the variation within a species (particularly when dealing with closely related species where there may only be approx. 1% difference between the species but maybe 0.5% difference within a species (ideally extensive population data would ensure markers chosen for ID can resolve differences between closely related species). It would have been useful to include intraspecies variation within Table S2 (on the diagonal) to give a fuller picture of variation within COI or if the data are not available at least give some indication in the text of the general intraspecies variation seen in COI.

Answer: Thank you very much for your valuable comments. The sentences have been added at the end of the second paragraph in Discussion. “However, the sequence data is based on a single specimen for each species and as such the accuracy of estimates of inter species variation are limited. The validity of a genetic marker depends not only on the interspecific variation but also the intraspecific variation (particularly when dealing with closely related species where there may only be approximate 1% difference between the species but maybe 0.5% difference within a species. Therefore, further investigation to reinforce the suitability of this genetic marker in more species is required, particularly in the closely related species.”

  1. The HRM analysis is based on 3 specimens, it is still unclear how reproducible the peak Tms are. Is the value reported an average temperature from the three specimens if so the standard deviation around this would be important especially as some Tms for different species are quite close, showing the variation in measurements would add support to this being a novel and appropriate method of species ID. It would also be useful to understand the sequence divergence across this 300 bp target (i.e. how many nucleotide differences are causing this change in Tm).

Answer: Your advice is really valuable. The HRM analysis is based on three specimens. Three replicates were performed for each sample. In this study, eight dominant species were selected for HRM analysis. A single characteristic peak was obtained for all species except L. porphyrina, and the Tm values of different species were separated. The peak value was 79.62 for H. ligurriens, 76.83 for S. peregrina, 73.35 for C. megacephala, 76.60 for S. dux, 76.35 for S. misera, 77.23 for S. sericea, and 73.13 for C. pinguis, and the corresponding Tm values were 73.15, 76.1, and 79.01 for L. porphyrina, respectively (Figures 4-5). The reported temperature values are from a single sample, not from the average of three samples. The results of the other two experiments are not presented in the paper. However, we found that the Tm value of two peaks is less than 0.15 °C in three repeated experiments, indicating that if the Tm value between species exceeds 0.15 °C, it can be identified as two different DNA sequences. The length of amplified product was 278 bp in this study.

Minor edits/typos:

  1. Line 52/53 - you say insect development is species specific, more specifically it it the timing of each life stage at any given temperature that is species specific which is why correctly species identification is so important. This could be more clearly stated here.

Answer: Thank you very much for your valuable comments. The sentences have been added at the first paragraph in Introduction. “but insect development is species-specific [3,4], more specifically, the timing of each life stage at any given temperature is species specific, which is why correctly species identification is so important. Additionally, there are significantly differences in the geographical distribution of insects. Accurate species identification and up-to-date locality information are essential for the effective application of forensic entomology in criminal investigations [5].”

Line 54 - Change "Whilst" to As there are significant differences in the geographical......

Answer: “Whilst” has been changed into “As”.

  1. Good to see the Genbank accession numbers for the specimen sequences but this may be better placed in the supplementary data (could just refer to OQ519752 - 81 in the main text).

Answer: This part has been modified. “Based on the above results, COI sequences of the common necrophagous flies on Hainan Island were submitted to Genbank (Accession no. OQ519752- OQ519781) (Table 2).”

  1. Supplementary data - not sure Figure S1 really adds anything, Table S2 I think would be better placed in the main body of the manuscript (add the length of COI sequence in the figure legend- is it the full length gene or just an amplified section?).

Answer: Figure S1 has been deleted. Table S2 should be presented in the manuscript, but there are too many data in this Table, considering that it is not easy to typeset, so it is placed in the supplementary file. The length of COI sequence for all species was 1,299 bp, which was added in the Table 2.

All the authors of this paper have agreed the finally revised manuscript. We wish to take this opportunity to thank you for consideration of publishing our paper.

Once again, thank you for your suggestions and comments.

Reviewer 3 Report (Previous Reviewer 3)

Comments and Suggestions for Authors

Fairly well polished this time, but I have proposed a few typographical alterations (all highlighted in green in the attached file), after which the manuscript should be ready to proceed to publication. 

Author Response

Reviewer 3

Fairly well polished this time, but I have proposed a few typographical alterations (all highlighted in green in the attached file), after which the manuscript should be ready to proceed to publication.

Answer: Thank you very much for your recognition of our study. The comments were extremely helpful to improve the quality of the manuscript. We have modified all the suspicious problems in revised manuscript. The revised parts have been highlighted and edited with track changes.

It is not clear how these two numbers relate or why they are different or if referring to the same specimens, why the sentence is duplicated. “According to the morphological classification of necrophagous flies, a total of 12,251 flies were obtained from 5 families, 17 genera, and 36 species in Hainan Island, among which 12 species were more than 200 specimens. These species were Chrysomya megacephala (3,400), Sarcophaga peregrina (1,346), Chrysomya rufifacies (1,131), Sarcophaga misera (1,008), Hemipyrellia ligurriens (980), Sarcophaga sericea (700), Sarcophaga cinerea (630), Sarcophaga dux (562), Chrysomya pinguis (536), Musca domestica (416), Lucilia porphyrina (205), and Chrysomya villeneuvi (200). A total of 11,114 specimens were captured, accounting for 90.72% of the total samples captured in Hainan Island (Table S1). ”

Answer: The sentences have been modified. “According to the morphological classification of necrophagous flies, a total of 12,251 flies were obtained from 5 families, 17 genera, and 36 species in Hainan Island, among which 12 species were relatively common. These species were Chrysomya megacephala (3,400),…….and Chrysomya villeneuvi (200), a total of 11,114 specimens were captured, which accounted for 90.72% of all the samples captured in Hainan Island (Table S1).”

In Figs 5-6, Why is the control colour for the line different to that used in the legend?

Answer: We apologize for this lack of seriousness. the control color has been modified.

All the authors of this paper have agreed the finally revised manuscript. We wish to take this opportunity to thank you for consideration of publishing our paper.

Once again, thank you for your suggestions and comments.

This manuscript is a resubmission of an earlier submission. The following is a list of the peer review reports and author responses from that submission.

Round 1

Reviewer 1 Report

Comments and Suggestions for Authors

The manuscript “Geographical Distribution and multimethod species identification of forensically Important Necrophagous Flies on Hainan Island” presents new monitoring and molecular data of necrophagous flies that were sampled using traps. Despite the authors' efforts, the manuscript does not meet the standards of this journal, even though it deals with an important topic. From a long list of issues with this manuscript, here are a few reasons that led to my decision:

·         The manuscript is poorly written especially in important sections such as material and methods, results, and discussion. The authors work is hard to understand, especially what they want to convey and why this study is valuable to forensic scientists.

·         Many important details in the description of the sampling design are missing such as:

o   The type of traps

o   Number of sampling days

o   Dates of the sampling days etc.

o   Statistical analysis

·         The presentation of the results is nowhere near the quality to be published. All Figures are in bad quality; species names are written like “Luciliacuprina” instead of “Lucilia cuprina”; and I don’t understand why the authors did not simply prepare a species list in a table.

·         In my opinion, the authors have no or low experience in the field of forensic entomology, which is evident in every part of this study. Here are some examples that illustrate this:

o   Forensic entomology is a key word and is also used in the text but the concept and the techniques of this discipline are not described at all!

o   Furthermore, the word “PMImin”, which is a key in this discipline of forensic science is missing at all.

o   There is also no explanation on how forensic entomology works and why the current study is important for forensic entomological case work

·         The authors should focus more on the new technique of species identification rather than trying to analyze the geographical distribution of necrophagous flies. I encourage the authors to revise the manuscript and writing a technical note on HRM curve analysis and not an original article.

·         The discussion of this manuscript is also problematic, as the authors are not able to really discuss their results, especially how their data can be used in forensic entomology. Sentences like „it is of great significance to study the community structure of necrophagous flies in this region in order to estimate death sites” shows clearly that the authors did not understand the concept of forensic entomology, or how to use and analyze monitoring data and especially their data. How do the authors want to use their data to estimate death sites? In my opinion this is not possible at all no matter how good the monitoring data of a study are.

Overall, the manuscript needs to be revised, and I recommend the authors to talk to an expert in the field of forensic entomology to discuss their results and improve their manuscript. In addition, it is more promising to focus on the new molecular method and write a methodological publication.

I have several questions and comments for the authors directly added to the PDF.

Comments on the Quality of English Language

The english must be corrected by a native speaker

Reviewer 2 Report

Comments and Suggestions for Authors

The paper covers the identification of key fly species of forensic importance on the island of Hainan and as such provides useful information for the region as well investigating an alternative approach to species identification in terms of high resolution melt curve analysis, which could be a quick and cheaper method of molecular identification. The authors present a clear assessment of the issues encountered in entomological identification and the potential benefits of this new approach.

Specific issues to be addressed:

Section 2.2 - it would be useful to included details of the PCR reagent composition (reagent concentration/volumes). Whilst the authors state that 30 species were used, it is not clear how many individuals of each species were sequenced.

Section 2.3 – sample sizes needed, how many of each of the 8 key species were tested for the HRM analysis.

Section 3.2 (Lines 264-272) -  the authors present data on the level of interspecies variation which supports the ability of COI to differentiate between species but it would be useful to see this in comparison to the intraspecies variation as this would lend greater support to COI as an appropriate target.

Section 3.3 – it would be useful to have data on how reproducible these Tms are for the HRM analysis as well as an indication as to how many SNP sites differ between these species. L. porphrina melt curves presented as three peaks, is this an issue with optimisation of the PCR in this species leading to non-specific amplification or does this reflect intraspecies variation. The blind testing of an unknown is a useful indication that this may be a useful approach but much further validation is needed.

Minor points:

Figures a little small, image quality/size needs to allow text to be clear

Line 72 should be bp not pb for the length of DNA sequence

Lines 122/3 and 217 there seems to be a repeat of the phrases semi-arid and semi-humid

Figure 1 – in the legend it would be useful to direct readers to table 1 for details about the different location whose images are in figure 1

The authors talk about creating a Genbank database -  this would be better phrased as sequences were submitted to Genbank

Reviewer 3 Report

Comments and Suggestions for Authors

This is a most interesting and well researched paper. There are some minor typological issues to correct - these are marked in the accompanying file by red text and the page line highlighted in yellow. A number of instances where red font colour is used requires to be changed to black font colour.

Besides this, my only real concern is that the nomenclature of fly names is not consistent between text and figure legends or axis labels in plots. Please be sure to use consistent names throughout the text and figures so that confusion is avoided. For example, Sarcophaga cinerea is referred to in the text, yet in the figures it is referred to as Leucomyia cinerea.

Please also ensure that generic and specific names are separated by a space, even in the axis of plots, otherwise it is near impossible to quickly pick out species.

Parts of the discussion are too long and could be broken into short paragraphs, but otherwise this clearly written and an interesting read.

Comments on the Quality of English Language